# Exploring the awe-some: Mobile eye-tracking insights into awe in a science museum

**Sheila Krogh-Jespersen**[1]*, **Kimberly A. Quinn**[2], **William L. D. Krenzer**[2,3],
**Christine Nguyen**[2], **Jana Greenslit**[4], **C. Aaron Price**[4]

**1** Department of Medical Social Sciences, Institute for Innovations in Developmental Sciences, Feinberg School of Medicine, Northwestern University, Chicago, IL, United States of America, **2** Department of Psychology, DePaul University, Chicago, IL, United States of America, **3** The Duke Initiative for Science & Society, Duke University, Durham, NC, United States of America, **4** Museum of Science and Industry, Chicago, IL, United States of America

* sheilakj@northwestern.edu

**Data Availability Statement:** Data are available from the Open Science Framework: 10.17605/OSF. IO/V7CSY.

## Abstract

Informal learning environments provide the opportunity to study guests' experiences as they engage with exhibits specifically designed to invoke the emotional experience of awe. The current paper presents insight gained by using both traditional survey measures and innovative mobile eye-tracking technology to examine guests' experiences of awe in a science museum. We present results for guests' visual attention in two exhibit spaces, one chosen for its potential to evoke positive awe and one for negative awe, and examine associations between visual attention and survey responses with regard to different facets of awe. In this exploratory study, we find relationships between how guests attend to features within an exhibit space (e.g., signage) and their feelings of awe. We discuss implications of using both methods concurrently to shed new light on exhibit design, and more generally for working in transdisciplinary multimethod teams to move scientific knowledge and application forward.

## Introduction

Informal learning institutions (e.g., museums, zoos, and aquariums) are built to provide a learning context that is both informative and enjoyable: a mixture of cognition and emotion that presents the unique opportunity for studying their interplay. These environments afford a window into dynamic, multidimensional, in-the-moment experiences that lab environments often aim to mimic. Indeed, experiences may unfold rapidly; for example, research [1, 2] has shown that guests at art museums spend less than 40 seconds on average viewing artwork, whereas lab-based study of attention to artwork averaged less than 3 seconds (as cited by [1]). Examining real-world experiences is particularly relevant when studying ephemeral experiences that may be difficult for participants to report and require a high degree of manipulation from the laboratory environment to induce, specifically here, emotion. The current study examines subjective experiences of awe among guests at the Museum of Science and Industry, Chicago (MSI–Chicago), using traditional survey-based measures partnered with innovative mobile eye-tracking technology. Our goal is to demonstrate how the conjunction of methods enriches insights into momentary, ephemeral emotions experienced in naturalistic settings.

**Funding:** The author(s) received no specific funding for this work.

**Competing interests:** The authors have declared that no competing interests exist.

## Museums and awe

Every year, 73 million guests visit science museums, zoos, aquariums and other science-themed cultural institutions in the United States [3]. The missions of the institutions vary widely, but almost all have one common theme: to invoke a sense of awe and curiosity and use those emotions to educate and inform [4]. Museums and their programming are often designed explicitly to induce awe and similar emotions in their guests. Museum architects often use the compression and expansion effect to induce awe by having guests walk down narrow corridors before entering into a large, immersive open space. Guests come to museums "with expectations to see wondrous things that they cannot see in their everyday lives" [5]. During wildlife experiences (e.g., zoos, aquariums, wildlife tourism), reinforcing a guest's sense of awe can evoke powerful and enduring memories [6]. The mechanism could be in awe's role as one of the epistemological emotions, which have been shown to enhance learning and shape long-term memories of museum visits [7–9].

Awe-inducing experiences are characterized by perceived vastness and a need for accommodation [10]. Vastness may be a matter of sheer physical size, but it may also reflect social importance, conceptual breadth, explanatory power, perceptual-sensory detail, or volume of unexpected information [11]. All of these experiences motivate us to accommodate or make sense of the experience—that is, to learn. Awe is assumed to arise when current experience cannot be accommodated within existing knowledge. Individuals experience awe when they are confronted with an unexpected event (unexpected because it is either entirely novel or highly inconsistent with experience) and when the resulting knowledge gap can only be reduced through cognitive accommodation. As a catalyst for the search for meaning, awe may thus provide significant untapped potential for promoting science, technology, engineering, and mathematics (STEM) learning in informal, public environments.

In our own research, we have taken theoretically derived conceptualizations of awe as a basis for developing a self-report measure of transient or situational awe (Situational Awe Scale (SAS); [12]). Hundreds of individuals, participating via online surveys or in the lab, have told us about their experiences with awe. They have described places or events that have over-whelmed them and the thoughts evoked by the experience, before rating these experiences according to theoretically derived descriptors that we provided to them. Across several studies, we have established that situational awe comprises four factors: positive awe characterized by feelings of *connection*, negative awe characterized by feelings of *oppression*, *physical reactions* such as chills and goosebumps, and feelings of *self-diminishment* in relation to the vastness of the world. Importantly, individuals' responses to the SAS are also sensitive to both conditions that we control in lab settings and features of real-world contexts (including museums and cultural institutions).

## Supplementing self-report data with online methods

Traditional self-report measures of emotion shed light on the subjective experience of the guest, yet leave open the question of what aspects of the environment influenced that experience. To push the needle forward on our understanding, traditional measures should be complemented with innovation; therefore, in the current study we used surveys to measure experiences of awe during guest visits to a science museum and paired these self-report measures with mobile eye-tracking to measure guests' visual attention as they moved through the museum space. Guests wore lightweight glasses equipped with a video camera that filmed the environment in front of them while infrared sensors tracked their pupil movement (Tobii Pro Glasses 2; see Fig 1), enabling us to assess what features of the exhibit captured their visual attention. This technology affords guests with freedom of movement and independence; as

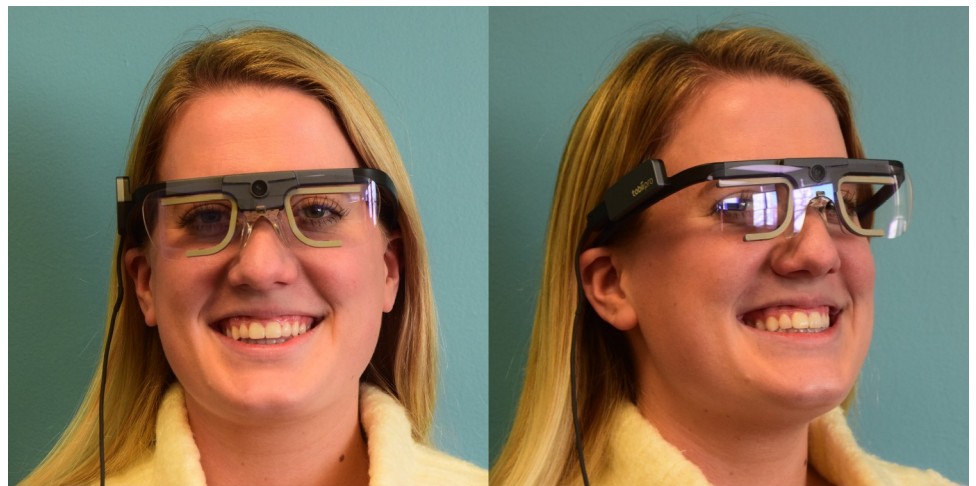

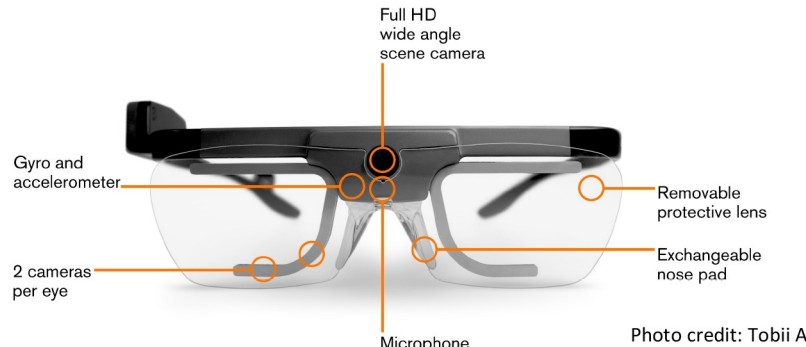

**Fig 1. Tobii Pro Glasses 2.** An example photo of a person wearing the Tobii Pro Glasses 2 (www.tobiipro.com) and schematic of the Glasses 2 with info. The individual pictured in Fig 1. has provided written informed consent (as outlined in PLOS consent form) to publish their image alongside the manuscript.

such, this mobile eye-tracking set-up is ideal for use in naturalistic spaces as it allows for in-the-moment data collection of what guests are attending to in an exhibit space [13–15] and is mostly unobtrusive as guests quickly become used to wearing video-recording devices [16]. Studies examining the feasibility of collecting eye-tracking data in museum settings have shown that most participants do not think that wearing this type of equipment interferes with their experiences [17]. Anecdotally, we have witnessed guests see themselves in a reflection and look surprised when they see the camera on their heads. Mobile eye-tracking affords similar levels of comfort, as we have viewed guests engage in behaviors, such as checking cell phone messages, that are typically considered more private.

Beyond its utility in facilitating naturalistic data collection, mobile eye-tracking also provides an online measure that is very likely to capture subtle influences on guest experience that guests are unaware of (or at least unable to report effectively). That is, guests might not be aware of subtle shifts in their own attention or how targets of attention influence their ongoing experience. Mobile eye-tracking, as an implicit online measure, thereby enhances the depth and reliability of our analysis. Indeed, research using this technology in science and art institutions is growing. For example, examining how people perceive artwork, researchers [18] used screen-based in-lab eye-tracking to examine whether gaze behavior was influenced by features of the paintings (e.g., dynamism and color). Moving this type of work "into the wild", a recent

study [19] used mobile eye-tracking to examine the gaze patterns of adults (n = 12) and children (n = 9) as they viewed a selection of paintings at the Van Gogh Museum in Amsterdam. This study included a manipulation of information regarding the paintings to examine whether this resulted in attentional differences. Findings revealed developmental differences in visual attention and ultimately supported the use of this type of method to provide new insight into how guests perceive artwork with visually salient features (e.g., brightness, hue).

As with self-report measures, of course, eye-tracking has limitations—in this case, providing no insight into subjective factors that either direct or result from attentional focus. A review [20] of issues related to mobile eye-tracking in museum environments in an exploratory case study with three participants ultimately concluded that for mobile eye-tracking to provide meaningful insights, it is best used in conjunction with other methods. Based on these informative reviews of mobile eye-tracking and its use in the field, the current study was designed to examine the emotional experience of guests in a science museum pairing traditional survey-based methods with innovative mobile eye-tracking technology. To our knowledge, this would be the first time mobile eye-tracking is utilized in to examine the psychological concept of awe within the real world setting of a science museum.

It is important to note that this is an exploratory, descriptive study. One issue with mobile eye-tracking is the intensive nature of the data collection and analysis, as participants' gaze typically has to be hand-coded; the coding algorithm cannot parse the regions of interest from truly dynamic and individual gaze recordings. (See [21] for a thorough review of issues related to mobile eye-tracking and design recommendations). The initial investigation that we report here was therefore directed toward collecting a small sample of responses that would enable us to assess the feasibility of this multimethod approach and its capacity to provide insights beyond unimethod approaches. Because of the limited sample size, we do not formally report any inferential statistics. Our goal, instead, is to argue for the methodological benefit of supplementing traditional self-report measures with eye-tracking technology. These two methods provide different information; neither on its own provides a comprehensive account of the phenomenon of interest, but the data from each approach serve to contextualize and clarify the other.

## Materials and methods

### Research setting

This study was conducted at the Museum of Science and Industry, Chicago (MSI–Chicago), located in Chicago, IL (USA). The MSI–Chicago has more than 400,000 sq. feet of exhibit space, with over 1.5 million guests visiting per year (2018). Given its size, the current study focused on two exhibit spaces that were chosen based on characteristics that were likely to influence guests' experience of awe: the Rotunda and the U-505 Exhibit (see Fig 2). Importantly, both spaces are physically vast, a factor known to induce awe [10], yet each space has unique features. In this paper we refer to study participants as guests instead of visitors, aligning ourselves with the preferred nomenclature used by our host institution.

The Rotunda was chosen given its likelihood to induce positive awe, as its features are vast, beautiful, and hopeful. The Rotunda is in the center of the museum and is among the first spaces most guests visit when they arrive. It is a tall, domed hall in a beaux-arts style with the words "Science discerns the laws of nature, industry applies it to the needs of Man" circling the bottom of the dome. It is aesthetically beautiful, but its vastness is further accentuated by the fact that guests approach it by taking an escalator or stairs up to it from a lower level; as they look up from the escalator, the dome looms into view high above them. This provides a physical sense of uplifting and welcomes them to the beginning of the MSI–Chicago experience.

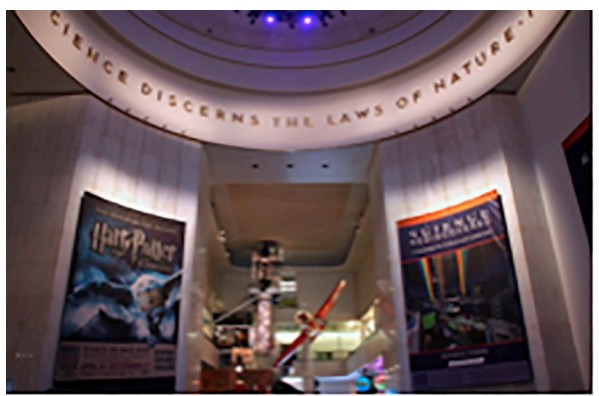
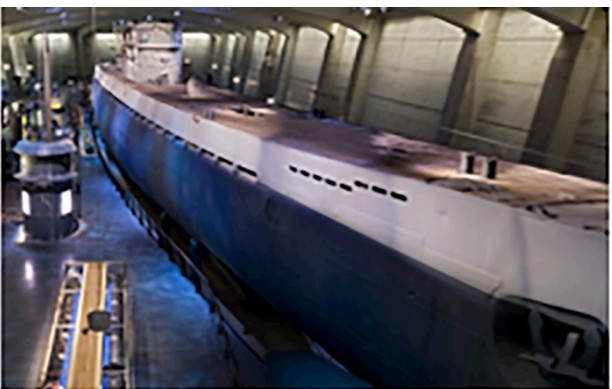

Rotunda
Positively valenced awe
Aesthetic beauty/uplifting

U-505 Exhibit
Negatively valenced awe
Historical significance/oppressive

**Fig 2. Still images taken to represent the vastness of the two selected exhibit spaces.** Image Sources: Rotunda: https://www.flickr.com/photos/mr_t_in_dc/3916523083; U-505: https://commons.wikimedia.org/wiki/File:U-505chicago.jpg.

The U-505 *Unterseeboot*, or U-boat (submarine), exhibit was chosen based on its likelihood of inducing negative awe. Guests enter the U-boat gallery after walking down a darkening, sinking, narrowing pathway (symbolically representing a submerging submarine) that includes description and images of the events leading to the real-life capture of the German U-boat by Allied forces in 1944. The audio soundtrack playing as guests walk down the tunnel includes gunshots, explosions that shake the floor and men's shouting. This dark tunnel makes a sharp left turn and then opens up to over-sized gallery where guests come face-to-face with the 700-ton, 252-foot U-505. Guests can then move around the U-boat on a raised platform that features informational signs. Dramatic lighting that is designed to mimic being under the Atlantic Ocean within the gallery and the ability to walk the length of the U-Boat add to the awe-inducing experience. The platform descends into an exhibit gallery with interactive simulations, videos, and relics from the submarine. Narrative content focuses on the drama surrounding the capture and the challenging daily lives of the sailors who served on board.

### Participants and recruitment

A total of 31 guests at the MSI–Chicago participated in the current study (18 female, 12 male, 1 undisclosed; $M_{age}$ = 21.67 years, SD = 6.96). In 2017, the MSI–Chicago instituted a policy of not collecting demographic information about race/ethnicity unless it is fundamental to the research question. However, a separate 2018 internal study established that this museum's guest population self-identifies as 64% White, 12% Latinx, 12% Asian and Asian ethnicities, 8% Black or African American, 1% American Indian or Alaskan Native, 1% Middle Eastern or North African, 1% Native Hawaiian or Other Pacific Islander and 1% other race or origins. This research study was approved by the MSI-Chicago's Institutional Review Board (IRB).

Recruitment occurred prior to entry into target exhibit spaces: A total of 15 guests participated in this study within the Rotunda space, and 16 guests participated within the U-505 exhibit space. This was a between-subjects study design such that a guest could only participate in this study at one exhibit space. Given that attention to the exhibit was a variable of interest, we avoided approaching families with small children and groups of four or more individuals. Also, given that guests were asked to wear Tobii Pro Glasses 2, we avoided approaching guests who were already wearing eyeglasses. Participants were thanked for their participation and given a gift certificate

for the museum's gift store. This study was approved by the MSI–Chicago's Institutional Review Board and all participants participated in an informed consent procedure.

## Procedure

Guests were recruited either as they approached the escalators leading up to the entrance of the Rotunda or as they approached the exhibit leading them down the hall to the gallery containing the U-505. Instructions for guests were similar for both exhibits, and they were recruited before they saw any part of the given exhibits. Once informed written consent was given, guests were guided through the Tobii Pro Glasses fitting procedure in which the Tobii Pro Glasses were placed and secured on their faces and the recording unit was secured to their clothing. Researchers [22] have noted issues with data quality when eye-tracker slippage occurs (e.g., participant removes or adjusts the Glasses on their face); we provided instructions to avoid these occurrences and guests were monitored as they were wearing the Glasses in the exhibit space. This fitting was followed by a fast calibration period in which guests focus their gaze on a bulls-eye pattern at arms-length while wearing the Tobii Pro Glasses. Once eye-to-glasses calibration was successful, guests were instructed to explore the exhibit space at their own pace. They were not given directive information regarding where they should attend or what information would be relevant. Guests took as much time as they preferred and were told to proceed as they normally would with the other guests (if any) in their party. A research assistant followed behind with a tablet that provides a real-time visual and audio representation of what the guest was viewing; this was only viewed for data collection monitoring and to troubleshoot any issues with the wearability of the Glasses. Research assistants maintained a distance that allowed for visual contact with the guests but not to disrupt the guests' experience. Recordings from the Tobii Pro Glasses were ended when the guest left the Rotunda space or when the guests rounded the completed a walk around one half of the U-505.

Guests completed a demographic questionnaire (age and gender) and the 18-item Situational Awe Scale (SAS; [12]) primarily on tablet devices, although paper surveys were available if requested. The SAS was designed to measure all aspects of awe, including some of its negative connotations. Its 18 items reflect four facets of awe: connection (e.g., "I felt psychologically connected to everyone/everything around me"), oppression (e.g., "Everything seemed disjointed"), chills (e.g., "I felt chills), and diminished sense of self (e.g., "I felt like I was trivial in the grand scheme of things"). Participants rated the extent to which they disagreed versus agreed with each statement as it applied to how they felt while walking through the target location, along a 7-point scale anchored by -3 (*disagree strongly*) and +3 (*agree strongly*); item order was randomized.

## Apparatus

The Tobii Pro Glasses 2 (see Fig 1) are equipped with four eye cameras, a gyroscope, and accelerometer and sample at 100Hz, while only weighing approx. 1.6 oz. The Glasses are attached via a cord to a recording unit that clips to the guests' clothing, enabling guest independence and freedom of movement. The recording unit includes a memory card that saves the guests' gaze data onto a video of their environment and an audio recording of what they say and hear. This memory card is later inserted into a computer with Tobii Pro Lab Analyzer to import the data for processing and analysis.

## Coding

Photographs that represented the exhibit spaces were taken to include design features, for example, the informational signs within the U-505 exhibit as well as the U-505 itself. Eight

photographs were taken to represent the views of the U-505 as a person moves through the exhibit space, and 7 of those photographs also included the information signs. An additional 3 photographs were taken of information signs that did not include the U-505 in the image. In the Rotunda, 3 photographs were taken to represent the following: the ceiling feature; the Science quote etched into the dome of the Rotunda; the large light fixture in the middle of the dome; and the two giant posters advertising present/upcoming exhibits within the MSI. The Rotunda also offers a view of a large exhibit just beyond the Rotunda that features an exhibit referred to as the Coal Mine, which includes a large hoist, and full-size airplanes hanging from the ceiling. Attention was coded to this view as well.

Although Tobii Pro Lab has an automatic coding feature, the nature of the study, which allowed uninhibited exploration through exhibits with varying participant movement through the space, other individuals in the scene, and variations in scene dynamics made this automatic mapping unreliable. Therefore, manual coding in Tobii Pro Lab frame-by-frame from the participant videos was conducted using the photographs from the exhibit spaces.

## Results

Data from the Tobii Pro Glasses 2 were exported and analyzed in Tobii Pro Lab Analyzer using the Tobii Attention Fixation Filter. To assess guests' visual attention, still photographs were taken from various angles of the two exhibit spaces that encompassed guests' visual range. Then, trained research assistants (one who served as the primary coder and one who served as a reliability coder) hand-coded each guests' visual patterns onto the still images to allow for analyses across participants as to what features of the exhibit captured their visual attention (see Coding section for more detail). For the current study, fixations were categorized based on salient features of the exhibit space by creating Areas of Interest (AOIs). Visual fixations were coded relative to these AOIs and data are presented as raw time and proportions. For both exhibit spaces, visual fixations to other features of the environment were excluded from analyses (e.g., fixations to other people and phones) given the focus on the features of the exhibit spaces themselves and not the general environment.

Visual attention in the Rotunda was analyzed according to five AOIs: the ceiling dome, the dominant light fixture in the center of the dome, the "Science Discerns the Laws of Nature. . .Industry Applies Them to the Needs of Man" quote that circled the bottom of the dome, a combined poster AOI to represent the left and right posters that flanked the space, and the center exhibit observable upon approach to the Rotunda (a coal mine tower and airplane). Guests spent between 1 minute 5 seconds to 4 minutes 23 seconds in this space ($M$ = 1 minute 49 seconds; SD = 50 seconds).

Time attending to the five AOIs in this space were low, with fixation durations ranging on average from 3.34 seconds to the center exhibit space and 1.91 seconds to the posters. The remaining three AOIs reflected attention to the upper features of the Rotunda, including the ceiling dome ($M$ = 420 milliseconds), the Science quote ($M$ = 810 milliseconds), and the light fixture ($M$ = 350 milliseconds). Proportions of time spent attending to the relevant areas of interest were created using the time spent in the specific AOI divided by the total time spent attending to all relevant exhibit space AOIs. Given the low rates of looking to the three AOIs on the upper part of the Rotunda (i.e., the ceiling, the Science quote, and the light fixture), these were combined for the proportional analysis. On average, the distribution of the guests' visual attention was to the center exhibition space (44%), the two posters (33%), and the features presented on the dome area of the Rotunda (23%). Individual rates of visual attention to the features of the Rotunda are reported in Table 1.

**Table 1. Total time in seconds that participants attended to various key features of the Rotunda.**

| Participant | Center Exhibition Space | Combined Posters | Light Fixture | Science Quote | Ceiling Dome | Total Recording Duration |
|---|---|---|---|---|---|---|
| Participant001 | 0.74 | 5.16 | 0.36 | 0.00 | 0.60 | 263.39 |
| Participant002 | 9.86 | 3.32 | 0.24 | 0.06 | 0.38 | 91.64 |
| Participant003 | 1.92 | 0.96 | 1.10 | 1.56 | 1.94 | 65.12 |
| Participant004 | 0.94 | 0.40 | 0.34 | 0.20 | 0 | 101.54 |
| Participant005 | 3.42 | 2.28 | 0.00 | 2.74 | 0 | 141.58 |
| Participant006 | 2.18 | 0.38 | 1.14 | 1.26 | 0.74 | 129.84 |
| Participant007 | 0.28 | 2.32 | 0.00 | 0.08 | 0 | 131.35 |
| Participant008 | 4.18 | 3.18 | 0.44 | 2.96 | 0.46 | 73.46 |
| Participant009 | 0.40 | 2.00 | 0.00 | 0.00 | 0 | 76.70 |
| Participant010 | 5.66 | 1.44 | 0.52 | 0.78 | 0.82 | 141.99 |
| Participant011 | 0.56 | 0.22 | 0.00 | 0.26 | 0.26 | 73.63 |
| Participant012 | 3.60 | 4.68 | 0.62 | 2.18 | 0.38 | 88.63 |
| Participant013 | 1.26 | 0.72 | 0.24 | 0.00 | 0.06 | 91.36 |
| Participant014 | 5.30 | 0.80 | 0.20 | 0.00 | 0 | 74.25 |
| Participant015 | 9.75 | 0.76 | 0.00 | 0.08 | 0.64 | 87.94 |
| Average | 3.34 | 1.91 | 0.35 | 0.81 | 0.42 | 108.83 |
| Standard Deviation | 3.16 | 1.58 | 0.37 | 1.07 | 0.51 | 50.10 |

For the U-505 exhibit space, 10 still images were used to categorize visual fixations within the large U-505 gallery with a total of 8 AOIs that represented the U-505 boat, and 11 informational signs about the U-505. Guests spent between 4 minutes 43 seconds to 23 minutes 34 seconds exploring the exhibit (*M* = 11 minutes 8 seconds; *SD* = 5 minutes 25 seconds). Data were combined across the 11 informational signs given variation in looking behaviors ranged due to a number of factors, including external factors such as whether the signs were blocked by other guests. On average, guests spent 1 minute 37 seconds visually attending to the informational signs and 32.25 seconds attending to the U-505. Again, these numbers are surprisingly low given the total amount of time spent in the exhibit space, but our strict criteria requires a direct fixation onto the AOI of interest to be counted (see Fig 3 for an example of gaze data to the U-boat). With regard to proportional analysis, guests spent 64% of their time attending to

Panel A

Panel B

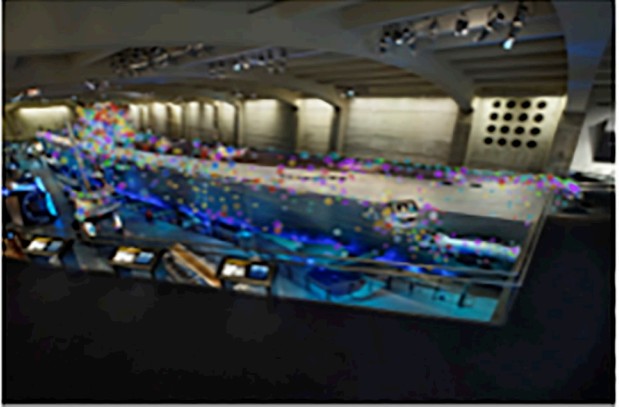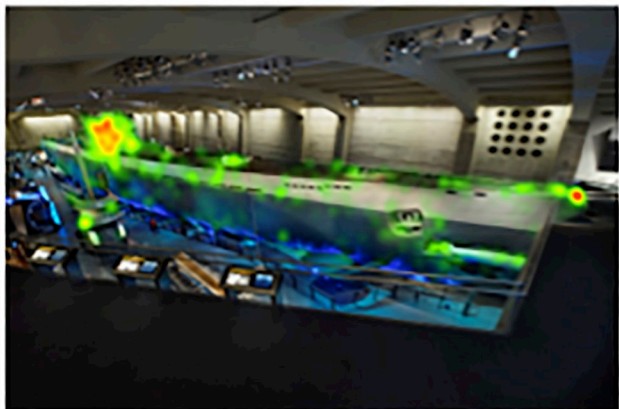

**Fig 3. A visual summary of guests' attention to the U-505 at the MSI–Chicago.** In Panel A, each color represents a different guest's individual visual pattern of attention across his/her visit. The circles each represent a fixation, meaning the guest focused his/her visual attention to that place, and the size of the circle represents the length of time that the guest attended to that place. Panel B presents heat maps representing the density of visual attention across guests to features of the U-505.

| AOI Images* | U505 Front | U505 | Sign 1 | Sign 2 | U505 | Sign 3 | U505 | Sign 4 | U505 | Sign 5 | U505 | Sign 6 | U505 | Sign A | Sign B | U505 | Sign C | Soldier Cut-out | Interact-ive Sign |
|---|---|---|---|---|---|---|---|---|---|---|---|---|---|---|---|---|---|---|---|
| Participant | | | | | | | | | | | | | | | | | | | |
| Participant 001 | 8.16 | 1.94 | 29.37 | 4.24 | 0.76 | 6.66 | 12.90 | 19.32 | 2.32 | 0.56 | 7.04 | 0.90 | 2.18 | 0.00 | 19.04 | 1.34 | 0.84 | .90 | 1.12 |
| Participant 002 | 8.58 | 3.24 | 2.28 | 0.88 | 0.74 | 0.64 | 0.96 | 0.82 | 0.96 | 0.66 | 4.80 | 0.16 | 1.54 | 0.00 | 37.75 | 0.38 | 5.44 | 2.04 | 0.42 |
| Participant 003 | 7.16 | 5.64 | 4.24 | 2.20 | 1.90 | 18.12 | 6.88 | 0.58 | 1.62 | 2.10 | 5.38 | 0.00 | 1.84 | 0.00 | 15.14 | 3.30 | 1.66 | 5.64 | 2.64 |
| Participant 004 | 1.40 | 0.92 | 21.44 | 2.18 | 13.48 | 0.30 | 0.90 | 0.00 | 6.94 | 10.24 | 2.08 | 7.74 | 11.64 | 2.88 | 32.05 | 6.34 | 2.78 | 5.54 | 24.80 |
| Participant 005 | 15.84 | 6.00 | 66.71 | 0.94 | 7.12 | 27.76 | 17.16 | 48.67 | 1.20 | 10.90 | 2.38 | 0.44 | 2.28 | 3.82 | 15.06 | 1.52 | 17.94 | 6.10 | 20.64 |
| Participant 006 | 10.82 | 3.88 | 1.56 | 1.72 | 7.18 | 2.38 | 5.02 | 2.44 | 0.98 | 1.84 | 3.06 | 0.00 | 12.02 | 54.47 | 16.54 | 3.16 | 17.12 | 98.36 | 39.15 |
| Participant 007 | 1.72 | 3.82 | 1.58 | 0.00 | 0.60 | 2.26 | 10.45 | 1.54 | 1.34 | 0.00 | 5.92 | 0.14 | 1.48 | 0.00 | 1.28 | 0.32 | 1.72 | 3.60 | 2.30 |
| Participant 008 | 2.38 | 9.46 | 0.00 | 18.98 | 6.38 | 36.95 | 8.78 | 26.06 | 5.22 | 30.89 | 4.80 | 42.91 | 0.54 | 1.64 | 22.24 | 0.00 | 12.46 | 2.10 | 29.48 |
| Participant 009 | 24.00 | 7.04 | 0.14 | 10.60 | 0.66 | 2.24 | 1.22 | 3.58 | 0.24 | 3.10 | 3.28 | 0.70 | 12.38 | 3.72 | 7.42 | 0.58 | 2.10 | 7.02 | 5.54 |
| Participant 010 | 2.62 | 1.26 | 0.32 | 1.90 | 0.92 | 0.00 | 0.44 | 0.00 | 0.58 | 0.90 | 3.74 | 0.20 | 6.08 | 7.08 | 1.28 | 1.24 | 2.98 | 4.24 | 1.64 |
| Participant 011 | 5.08 | 0.76 | 0.00 | 26.40 | 0.86 | 32.81 | 0.18 | 0.00 | 0.00 | 5.10 | 2.56 | 9.98 | 1.84 | 1.94 | 9.30 | 0.06 | 7.32 | 2.54 | 10.32 |
| Participant 012 | 0.36 | 0.96 | 0.00 | 47.19 | 0.36 | 4.10 | 12.14 | 28.90 | 0.24 | 5.32 | 0.40 | 4.40 | 1.62 | 8.18 | 48.61 | 0.00 | 0.44 | 18.54 | 19.00 |
| Participant 013 | 1.82 | 7.34 | 0.00 | 0.88 | 0 | 0 | 4.98 | 0.54 | 1.84 | 0.00 | 8.06 | 0.00 | 0.60 | 0.00 | 0 | 0.00 | 0.24 | 12.30 | 5.58 |
| Participant 014 | 1.94 | 6.42 | 7.16 | 1.80 | 0 | 0 | 7.64 | 0.00 | 4.32 | 0.00 | 9.86 | 46.63 | 3.80 | 1.66 | 0.24 | 4.84 | 0.00 | 1.66 | 114.68 |
| Participant 015 | 15.90 | 2.96 | 0.00 | 0.00 | 0 | 0 | 0.34 | 0.00 | 4.18 | 3.30 | 4.44 | 0.76 | 2.80 | 0.06 | 0 | 0.00 | 2.16 | 1.46 | 1.22 |
| Participant 016 | 2.00 | 7.44 | 0.12 | 0.58 | 13.86 | 0.00 | 0 | 0 | 0 | 0 | 6.10 | 0.00 | 0.58 | 0.00 | 0 | 0 | 0 | 3.08 | 5.50 |
| Average | 6.86 | 4.32 | 8.43 | 7.53 | 3.43 | 8.39 | 5.63 | 8.28 | 2.00 | 4.68 | 4.62 | 7.18 | 3.95 | 5.34 | 14.12 | 1.44 | 4.70 | 10.95 | 17.75 |
| Standard Deviation | 6.78 | 2.78 | 17.73 | 12.99 | 4.73 | 12.89 | 5.46 | 14.57 | 2.08 | 7.79 | 2.43 | 14.99 | 4.22 | 13.35 | 14.96 | 1.96 | 5.96 | 23.75 | 28.44 |

*Note that images and AOIs may overlap but gaze points were never double-coded across images.

**Fig 4. AOI images for the U-505 with individual participants' total time attending to key features of the U-505 exhibit.** Ten photographs present the 8 AOIs that represent the U-505, with attention calculated across these AOIs for a summed score, as well as the 11 AOIs that represent the informational signs present in the U-505 exhibit. Each participant's looking time is calculated in seconds.

information presented in signs and 36% attending to features of the U-boat. Individual rates of visual attention to the features of the U505 are reported in Fig 4.

Going beyond summary measures of visual attention, guests' responses to the SAS were explored according to the distributional pattern of their visual attention within each exhibit space. Proportions of attention were created by participant from their total time looking to the relevant AOIs within each exhibit space. Although the small samples preclude the use of inferential statistics to draw strong conclusions, plotting guests' visual attention scores against their SAS subscale scores yielded interesting patterns (see Figs 5 and 6). Because correlations are extremely likely to be unstable and thus unreliable with such small sample sizes (e.g., [23]), we do not report the correlation coefficients here. For the reader's interest, however, these results are depicted in Supplementary Materials (S1 and S2 Tables).

Although only exploratory, this analysis hints at the potential contribution of mobile eye-tracking to expanding our understanding of complex psychological processes. To the extent that guests in the Rotunda spent proportionally more time gazing upward at the ceiling—the feature of the Rotunda that seemed intuitively most likely to elicit awe—they reported more connection and a more diminished sense of self. Descriptively, the patterns also suggest that visual attention to other features of the space might undermine awe. Proportionally more time gazing beyond the

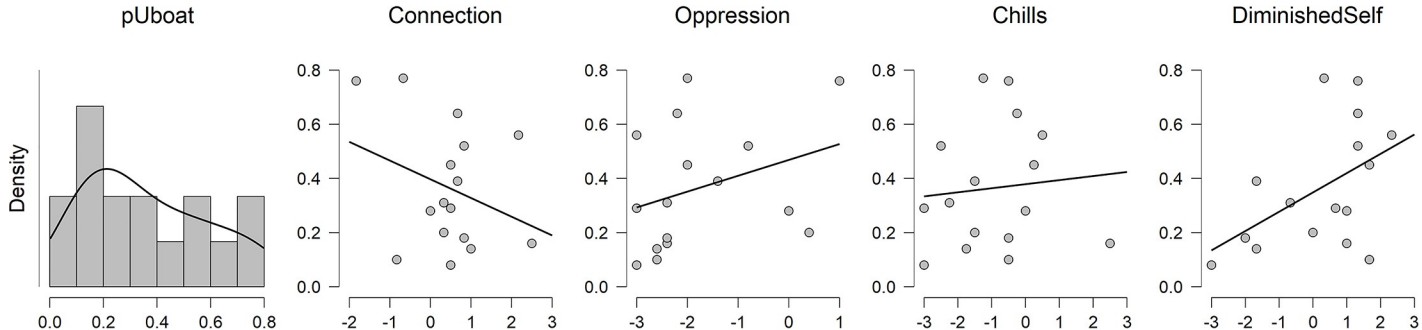

**Fig 5. Proportion of visual attention in the Rotunda.** Proportion of visual attention directed toward the central exhibits beyond the Rotunda, the posters flanking the entry to the exhibits, and the dome, light fixture, Science quote (referred to as Dome in this Fig) of the Rotunda as a function of SAS subscale scores.

**Fig 6. Proportion of visual attention in the U-505 exhibit.** Proportion of visual attention directed toward German U-Boat (versus the informational signage) as a function of SAS subscale scores. *Note.* We do not present the results in terms of the proportion of attention directed to the informational signage because this would simply be the reciprocal of the U-Boat results.

Rotunda to the more distant exhibits—away from the space they were in—was associated with weaker feelings of connection; proportionally more time gazing at the posters flanking the entrance to the other exhibits was associated with less of a diminished sense of self.

To the extent that guests in the U-505 exhibit spent proportionally more time looking at the German U-boat—whose size and significance seemed intuitively likely to elicit awe—they, like the guests in the Rotunda who focused on the dome, reported a diminished sense of self. Further, submarine-focused gaze in the U-505 exhibit was also associated with stronger feelings of oppression and *weaker* feelings of connection, a pattern that makes sense in the context of a negative awe experience.

## Discussion

The current exploration paired traditional survey measures with innovative mobile eye-tracking technology to examine the ephemeral experience of awe across two exhibit spaces within a science museum. Attention to features that required looking up into the dome of the Rotunda (i.e., the Science quote, the light fixture, and the ceiling dome) was descriptively related to greater feelings of connectedness and diminished sense of self. However, when attention was distracted away from the Rotunda to an adjacent exhibit space, guests reported a weaker sense of connection. This highlights a feature of exhibit design that may be critical to connection, the feeling of being in the moment rather than planning your next move. For example, a brief glance at a painting will provide the viewer with the gist of the painting but full appreciation of the artist's purpose requires attention to the details of the painting and the artist's strokes. These findings could help inform exhibit design to encourage a sense of connection by focusing guests' attention to the current environment, rather than hinting at what is to come.

In contrast, the darkness and mood of the U-505 exhibit space seemed to support feelings of negative awe, including greater levels of oppression and weaker sense of connection. Despite these negative feelings, guests spent more time in this exhibit on average than in the Rotunda. Interestingly, to the extent that guests in both the Rotunda and U-505 attended to aspects of the exhibits expected to elicit awe, they descriptively reported a diminished sense of self. An open question is whether this facet of awe underlies the sense of vastness that is often associated with experiencing awe.

The contribution of these data is to underscore how this multimethod approach—pairing traditional survey measures with innovative mobile eye-tracking technology—enables a deeper understanding of the phenomenon of interest. On its own, the survey data would speak to the question of what facets of awe characterize guests' subjective experiences in different museums exhibits (or at least what guests are able to self-report), but not which exhibit features are responsible for these experiences. Conversely, on its own, the eye-tracking data would speak to the question of what exhibit features capture and dominant guests' attention, but not whether or how attention matters for the subjective experience. Combining methods enables greater insight into the complexity of the phenomenon, identifying how cognitive processes interface with emotional experience. One limitation of the current method is that surveys were conducted upon exiting the exhibit space; greater insight into in-the-moment experiences of awe would be gained by sampling responses while guests were experiencing various parts of the exhibit itself. The possible combination of text message survey sampling and eye-tracking would be an exciting future direction to pinpoint moments of high vs. low awe within the exhibit space.

### Moving scientific knowledge and application forward via transdisciplinary field research

This exploratory project brought together a developmental scientist with expertise in eye-tracking, an experimental psychologist with interest in scale development and the psychology

of awe, and a learning scientist leading a research team in a major science museum. This collaboration provided fertile ground for cross-pollination of ideas and knowledge in ways that provided benefits to both scientists interested in gaining insight into psychological phenomena and science educators hoping to harness that insight.

Given the goals of public science institutions to educate and inform, understanding the circumstance that evoke awe may shed light on design and development opportunities to facilitate deeper learning. Laboratory evidence supports a role for awe in promoting critical thinking and learning, for example, when students are curious or experience interest—which are likely to accompany awe—they persist longer at learning tasks, spend more time studying, and get better grades [24]. Moreover, awe promotes critical thinking, in the form of a lower likelihood of being persuaded by weak arguments [25], promotes ethical decision-making [26], and increases skeptical thinking [27]. Our results elsewhere confirm that museum exhibits with different features elicit different profiles of awe [12] and that facets of awe experienced in a public zoo differ in their associations with beliefs about animals [28]. Our results here provide proof of concept that supplementing these traditional self-report measures with mobile eye-tracking technology clarifies the nature of these effects. If awe drives critical thinking and science museums create awe-inspiring experiences, then this multimethod approach could revolutionize how such experiences are designed.

From a scientific knowledge standpoint, collaborative field projects such as this one have the capacity to provide more ecological valid insights into ephemeral experiences such as the experience of awe. Although lab-based studies allow for a level of control that can isolate the distinct features drawing attention, museum-based studies provide insight into the actual experience in the moment of engaging with information that is designed to inspire curiosity and evoke awe. Research has shown that artwork viewed in a museum is better remembered than when viewed in a lab-based environment [29], suggesting (unsurprisingly) that experiences in naturalistic environments are qualitatively different from experiences in the lab. If our goal is to understand the relationships between visual attention and emotions like awe as they are naturally experienced, then they need to be studied in naturalistic settings. In studies such as the one reported here, bringing together expertise in visual attention, awe, and science learning with practical knowledge of how guests engage with museums enables the investigation of awe more authentically than in a lab setting.

Although it remains an empirical question as to whether the effects seen here are robust in a larger sample, the preliminary outcomes of this multimethod approach also point to potentially fruitful areas of future inquiry into the nature of the attention–awe relationship, with implications for our understanding of awe itself. In particular, beyond the suggestion that visual attention to awe-inducing stimuli is associated with self-reported connection for ostensibly positive awe elicitors (i.e., the dome of the Rotunda) and self-reported oppression for ostensibly negative awe elicitors (i.e., the U-505), the data also offer the tantalizing possibility that self-diminishment is fundamental to the experience. This is not a new suggestion (e.g., [27]), but our data provide the first hint that the feeling of self-diminishment is underpinned by visual attention.

## Conclusion

### The value of multimethod research

What exactly does the combination of traditional survey techniques and innovative mobile eye tracking add? Without the eye-tracking data, we could get a sense of how much awe guests experience in each location and perhaps even which dimensions of awe are activated in each location. This information is useful, but does not lend itself to clear recommendations for how to curate or design a better experience. With the eye-tracking data added, we get a more precise

picture. Our results suggest that visual attention to specific exhibit features has implications for the emotional experience. When we see differences in the extent to which awe is experienced, we have a tool that could enable us to determine why those differences emerge—which features heighten awe or which features inhibit it—and use these insights to inform design.

## Supporting information

**S1 Table. Correlations (and p-values) among measures, Rotunda.**
(DOCX)

**S2 Table. Correlations (and p-values) among measures, U505.**
(DOCX)

## Author Contributions

**Conceptualization:** Sheila Krogh-Jespersen, Kimberly A. Quinn, William L. D. Krenzer, Jana Greenslit, C. Aaron Price.

**Data curation:** Kimberly A. Quinn, William L. D. Krenzer, Christine Nguyen.

**Formal analysis:** Kimberly A. Quinn, William L. D. Krenzer.

**Investigation:** Sheila Krogh-Jespersen, Kimberly A. Quinn, C. Aaron Price.

**Methodology:** Sheila Krogh-Jespersen, Kimberly A. Quinn, William L. D. Krenzer, Christine Nguyen, C. Aaron Price.

**Project administration:** Kimberly A. Quinn, Jana Greenslit, C. Aaron Price.

**Resources:** Sheila Krogh-Jespersen, Jana Greenslit.

**Software:** Sheila Krogh-Jespersen, Kimberly A. Quinn, Christine Nguyen.

**Supervision:** Sheila Krogh-Jespersen, Kimberly A. Quinn, C. Aaron Price.

**Visualization:** Sheila Krogh-Jespersen, William L. D. Krenzer.

**Writing – original draft:** Sheila Krogh-Jespersen, Kimberly A. Quinn, C. Aaron Price.

**Writing – review & editing:** Sheila Krogh-Jespersen, Kimberly A. Quinn, William L. D. Krenzer, Christine Nguyen, Jana Greenslit, C. Aaron Price.

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
