## [Decision Letter · Decision Letter 0]

29 Apr 2020

PONE-D-20-00804

Exploring the Awe-some: Mobile eye-tracking insights into Awe in a science museum

PLOS ONE

Dear Dr. Krogh-Jespersen,

Thank you for submitting your manuscript to PLOS ONE. After careful consideration, we feel that it has merit but does not fully meet PLOS ONE’s publication criteria as it currently stands. Therefore, we invite you to submit a revised version of the manuscript that addresses the points raised during the review process.

Although both reviewers acknowledge the idea behind the study, both reviewers have several and partly complementary comments regarding the hypotheses and methodology. In addition, the second reviewer has several technical questions that the authors should address accordingly in a new iteration.

We would appreciate receiving your revised manuscript by Jun 13 2020 11:59PM. To enhance the reproducibility of your results, we recommend that if applicable you deposit your laboratory protocols in protocols.io, where a protocol can be assigned its own identifier (DOI) such that it can be cited independently in the future. For instructions see: http://journals.plos.org/plosone/s/submission-guidelines#loc-laboratory-protocols

We look forward to receiving your revised manuscript.

Kind regards,

Enkelejda Kasneci, Ph.D.

Academic Editor

PLOS ONE

Journal Requirements:

3. Please provide additional details regarding participant consent. In the ethics statement in the Methods and online submission information, please ensure that you have specified whether consent was informed

4. We note that Figures in your submission contain copyrighted images. All PLOS content is published under the Creative Commons Attribution License (CC BY 4.0), which means that the manuscript, images, and Supporting Information files will be freely available online, and any third party is permitted to access, download, copy, distribute, and use these materials in any way, even commercially, with proper attribution. For more information, see our copyright guidelines: http://journals.plos.org/plosone/s/licenses-and-copyright.

1.    You may seek permission from the original copyright holder of Figures to publish the content specifically under the CC BY 4.0 license.

5. We note that Figure 1 includes an image of a [patient / participant / in the study]. 

Additional Editor Comments (if provided):

Reviewers' comments:

Reviewer's Responses to Questions

**Comments to the Author**

1. Is the manuscript technically sound, and do the data support the conclusions?

Reviewer #1: Yes

Reviewer #2: Partly

2. Has the statistical analysis been performed appropriately and rigorously? 

Reviewer #1: N/A

Reviewer #2: N/A

3. Have the authors made all data underlying the findings in their manuscript fully available?

Reviewer #1: No

Reviewer #2: No

4. Is the manuscript presented in an intelligible fashion and written in standard English?

Reviewer #1: Yes

Reviewer #2: Yes

5. Review Comments to the Author

Reviewer #1: This is an interesting exploratory study combining survey and mobile eye tracking methods in a science center, a real-world-setting where informal learning takes place. In this context I very much appreciated reading about "awe" as this presents a clear and specific research focus that is of potential interest to informal learning institutions. Here, fascination often comes from feelings of wonder or sensation and barriers from feelings of intimidation or insecurity that, to my mind, both tackle the concept of "awe". Consequently, I also appreciated reading the part where "awe" is defined and that the authors transferred a standardized survey (SAS Situational Awe Scale) from the lab to the field. What I am missing is a better contextualisation of this research in the area of museum and visitors studies with a huge amount of studies that deal with exhibition experiences. In this sense I also found it a bit uncommon to read about "guests" and not about "visitors" and their experiences on site. With respect to other mobile eye tracking studies, I would have liked to have clarified in what sense the authors believe that this is "the first time mobile eye-tracking is utilized in this type of research".

An important argument of the text is based on the necessity of method combinations when working with mobile eye tracking following Eghbal-Azar & Widlok (2013). On a meta level the authors very much make clear that such a method combination is important also in this study. In addition, the authors demonstrate understandably the importance of working in a trans-disciplinary team. However, my main point of critique is that the presented results and their discussion do too little to empirically make a point on the importance of mixed method research designs with mobile eye tracking and their use in informal learning settings. This is to my mind, not a problem of the study per se that seems well planned and professionally conducted but a problem of data analysis that is not going into depth. On the one hand, this refers to the known problem of complex mobile tracking data analysis with the necessity of laborous manual work. On the other hand I have the impression that the researchers could have studied their data more thoroughly in order to present more detailed results. This would also enable a refined discussion how the phenomenon of "awe" can be studied through visual attention patterns and self reported emotions. To shed new light on exhibition display, as stated in the abstract, also more empirical results and more interpretative thoughts would be necessary. If the authors are able to base their arguments better in their own empirical work (also presenting more tables and figures) and discuss their results in depth (also in comparison with other studies), I am very much looking forward to see this text published. This would be another milestone in advancing the use of mobile eye tracking in informal learning institutions.

Reviewer #2: 1) L81: Regarding the unobtrusiveness claim, do the authors have any data to

support this claim? Some evidence towards this general direction for

head-mounted eye trackers is provided by [1] with a large sample size, although

that work use a different eye tracking system. Please expand on this item in the

text.

2) L175: Please clarify in the text: 2.1) did the authors switched between the

Tobii-provided nose pieces as necessary or a single size was used? 2.2)

did the authors employ the corrective lenses if necessary or all participants

reported sufficient visual acuity without glasses?

3) L210: Was each participant's data coded by a single coder or the same data

was coded by multiple coders to decrease subjectivity? This too should be clear

in the text.

4) Still regarding coding: Tobii provides a solution for automatic gaze mapping.

Did the authors tried using it instead of manual coding? If so, a brief

paragraph on their experience with the automatic gaze mapping would be

interesting even if a bad one (e.g., [2]).

5) When mapping to the still images, how did the coders dealt with gaze

positions? Was the gaze point translated to the exact matching position or to a

"nearest target"? This is relevant since [3] suggests that, for upward gaze

directions, Tobii Glasses 2 suffers from large deviations and the participants

spent more time gazing upwards (L261). Please clarify in the text.

6) L258: The authors mention "interesting patterns". Personally, I didn't notice

any correlation in Figs 5 and 6. I suggest the authors to clearly describe which

effects they have noticed in the text and, despite the reduced sample size,

conduct the appropriate tests even if they don't end up being statistically

significant.

7) L364: Please expand on which exhibit features had emotional experience

implications.

[1] Santini, Thiago, et al. "The art of pervasive eye tracking: unconstrained eye tracking in the Austrian Gallery Belvedere." Proceedings of the 7th workshop on pervasive eye tracking and mobile eye-based interaction. 2018.

[2] Herlitz, Mattias. "Analyzing the Tobii Real-world-mapping tool and improving its workflow using Random Forests." (2018).

[3] Niehorster, Diederick C., et al. "The impact of slippage on the data quality of head-worn eye trackers." Behavior Research Methods (2020): 1-21.

6. PLOS authors have the option to publish the peer review history of their article (what does this mean?). If published, this will include your full peer review and any attached files.

Reviewer #1: No

Reviewer #2: No

---

## [Author Response · Author response to Decision Letter 0]

18 Aug 2020

5. Review Comments to the Author

Reviewer #1: This is an interesting exploratory study combining survey and mobile eye tracking methods in a science center, a real-world-setting where informal learning takes place. In this context I very much appreciated reading about "awe" as this presents a clear and specific research focus that is of potential interest to informal learning institutions. Here, fascination often comes from feelings of wonder or sensation and barriers from feelings of intimidation or insecurity that, to my mind, both tackle the concept of "awe". Consequently, I also appreciated reading the part where "awe" is defined and that the authors transferred a standardized survey (SAS Situational Awe Scale) from the lab to the field. 

What I am missing is a better contextualisation of this research in the area of museum and visitors studies with a huge amount of studies that deal with exhibition experiences. In this sense I also found it a bit uncommon to read about "guests" and not about "visitors" and their experiences on site. 

We appreciate this feedback and have added additional literature and text to the section, “Museums and Awe” to bolster our discussion. We also include a statement in the Research Setting section addressing our use of the term “guest” in the place of “visitor”.

With respect to other mobile eye tracking studies, I would have liked to have clarified in what sense the authors believe that this is "the first time mobile eye-tracking is utilized in this type of research".

We appreciate the reviewer pointing out the broadness of this statement and have clarified on page 6 that we mean “to examine the psychological concept of awe within the real world setting of a science museum.“

An important argument of the text is based on the necessity of method combinations when working with mobile eye tracking following Eghbal-Azar & Widlok (2013). On a meta level the authors very much make clear that such a method combination is important also in this study. In addition, the authors demonstrate understandably the importance of working in a trans-disciplinary team. However, my main point of critique is that the presented results and their discussion do too little to empirically make a point on the importance of mixed method research designs with mobile eye tracking and their use in informal learning settings. This is to my mind, not a problem of the study per se that seems well planned and professionally conducted but a problem of data analysis that is not going into depth. On the one hand, this refers to the known problem of complex mobile tracking data analysis with the necessity of laborious manual work. On the other hand I have the impression that the researchers could have studied their data more thoroughly in order to present more detailed results. This would also enable a refined discussion how the phenomenon of "awe" can be studied through visual attention patterns and self reported emotions. To shed new light on exhibition display, as stated in the abstract, also more empirical results and more interpretative thoughts would be necessary. If the authors are able to base their arguments better in their own empirical work (also presenting more tables and figures) and discuss their results in depth (also in comparison with other studies), I am very much looking forward to see this text published. This would be another milestone in advancing the use of mobile eye tracking in informal learning institutions.

We appreciate the call from the reviewer for more information about the visual patterns and more interpretation. We have included 2 tables in the revised manuscript that provide the individual patterns of attention in seconds to various features within the two exhibit spaces. We have also included exploratory correlational analyses between the eye-tracking measures and the Awe measures in supplementary materials with a note in text regarding the nature of correlations conducted with small sample sizes. We have also provided more detail regarding how mobile eye-tracking coding was conducted to shed light on this method and some of the constraints mentioned above. 

Reviewer #2: 1) L81: Regarding the unobtrusiveness claim, do the authors have any data to support this claim? Some evidence towards this general direction for head-mounted eye trackers is provided by [1] with a large sample size, although that work use a different eye tracking system. Please expand on this item in the text.

We include additional citations regarding the use of recording devices worn on the heads of study participants as well as qualitative reports from our study team regarding participant behavior to support this claim. We did not explicitly ask participants about their comfort level with the eye-tracking Glasses beyond ensuring that they were comfortable during the calibration and set-up of the equipment. They were informed that they could remove the Glasses at any time if they experienced discomfort. We also include the recommended cites from this review to direct readers to articles addressing these issues more specifically. 

2) L175: Please clarify in the text: 2.1) did the authors switched between the Tobii-provided nose pieces as necessary or a single size was used? 2.2) did the authors employ the corrective lenses if necessary or all participants reported sufficient visual acuity without glasses?

We did not switch between nose pieces, and just used a single size. We excluded approaching visitors with glasses (now discussed in the manuscript) and did not use corrective lenses in the eye-tracking Glasses.

3) L210: Was each participant's data coded by a single coder or the same data was coded by multiple coders to decrease subjectivity? This too should be clear in the text.

We clarify in the methods section that two coders where involved in the process: a primary coder and a reliability coder. 

4) Still regarding coding: Tobii provides a solution for automatic gaze mapping. Did the authors tried using it instead of manual coding? If so, a brief paragraph on their experience with the automatic gaze mapping would be interesting even if a bad one (e.g., [2]).

We have clarified in a description in the Coding section the many factors that mainly are due to the dynamic and individual nature of each guest’s exploration of the exhibit space that rendered this automatic mapping as unreliable. 

5) When mapping to the still images, how did the coders dealt with gaze positions? Was the gaze point translated to the exact matching position or to a "nearest target"? This is relevant since [3] suggests that, for upward gaze directions, Tobii Glasses 2 suffers from large deviations and the participants spent more time gazing upwards (L261). Please clarify in the text.

Although we are not entirely sure what is meant by “nearest target” here, we assume that means adjusting the eye-tracking data points based on an assumption of where the person was looking, which we did not do. We matched the gaze points to static images of the exhibit spaces, with examples of images for the U-505 now presented in Table 2 with more detail about individual coding and analyses.

6) L258: The authors mention "interesting patterns". Personally, I didn't notice any correlation in Figs 5 and 6. I suggest the authors to clearly describe which effects they have noticed in the text and, despite the reduced sample size, conduct the appropriate tests even if they don't end up being statistically significant.

We have provided correlation tables to be presented in Supplementary Materials. Because correlations are extremely unstable at small sample sizes (e.g., Schönbrodt & Perugini, 2013), we do not wish to draw attention to the inferential statistics and away from the broader message in the main text, but are happy to provide the numbers for the reader’s interest.

7) L364: Please expand on which exhibit features had emotional experience implications.

We have revised the manuscript to include the individual patterns of total attention directed toward various features within the exhibit spaces; however, we note that self-report measures were only taken at the end of the guests’ experience within an exhibit space and therefore cannot be tied to an exact moment or feature within the exhibit. We have clarified this limitation within the revised manuscript.

[1] Santini, Thiago, et al. "The art of pervasive eye tracking: unconstrained eye tracking in the Austrian Gallery Belvedere." Proceedings of the 7th workshop on pervasive eye tracking and mobile eye-based interaction. 2018.

[2] Herlitz, Mattias. "Analyzing the Tobii Real-world-mapping tool and improving its workflow using Random Forests." (2018).=

[3] Niehorster, Diederick C., et al. "The impact of slippage on the data quality of head-worn eye trackers." Behavior Research Methods (2020): 1-21.

 We appreciate these suggested references and have included two in the revised manuscript. We have opted not to include the Herlitz reference as it is a master’s thesis examining the real-world mapping tool in Tobii which we did not utilize for the current study.

---

## [Editor Report · Decision Letter 1]

2 Sep 2020

Exploring the Awe-some: Mobile eye-tracking insights into Awe in a science museum

PONE-D-20-00804R1

Dear Dr. Krogh-Jespersen,

We’re pleased to inform you that your manuscript has been judged scientifically suitable for publication and will be formally accepted for publication once it meets all outstanding technical requirements.

Kind regards,

Enkelejda Kasneci, Ph.D.

Academic Editor

PLOS ONE
---

## [Editor Report · Acceptance letter]

9 Sep 2020

PONE-D-20-00804R1 

Exploring the Awe-some: Mobile eye-tracking insights into Awe in a science museum 

Dear Dr. Krogh-Jespersen:

I'm pleased to inform you that your manuscript has been deemed suitable for publication in PLOS ONE. Congratulations! Your manuscript is now with our production department. 

Kind regards, 

on behalf of

Dr. Enkelejda Kasneci 

Academic Editor

PLOS ONE